# Multidimensional analysis of register variation in English translations of *Shijing*

**Baohu Li**, **Guangwei Li***

School of Languages and Literature, University of South China, Hengyang, China

* hu0701002@126.com

## Abstract

This study employs Multidimensional Analysis (MDA) to compare the register of Arthur Waley's and Ezra Pound's translations of *Shijing*, and further explores the factors contributing to their differences. The key findings are as follows: (1) Waley's translation corresponds to the "involved persuasion" register, characterized by high interactivity and extensive informational elaboration. In contrast, Pound's translation aligns with the "general narrative exposition" register, emphasizing informativeness and narrativity; (2) The interactivity in Waley's translation is primarily driven using analytic negation, first-person pronouns, and modal verbs, while the elaboration is attributed to the frequent use of demonstrative pronouns. In contrast, Pound's translation exhibits strong informativeness due to the frequent use of nouns and prepositional phrases, while its narrativity is shaped by synthetic negation and public verbs; (3) Waley's approach prioritizes an accurate reflection of ancient Chinese society and the preservation of cultural heterogeneity. In contrast, Pound's translation focuses on didacticism, emotional energy, and precision. The differences in the translators' ideologies and poetic philosophies are identified as the primary factors accounting for the register variations in their translations.

## 1. Introduction

Chinese classical literature has a rich legacy, and classical poetry is among the earliest forms to be translated and studied. Its translations significantly influenced Western societies [1]. *Shijing* (诗经, *The Book of Songs*) is the oldest surviving anthology of ancient Chinese poetry. In 1736, Richard Brookes translated eight *Shijing* poems into English from French, such as *Tian Zuo* (天作) and *Huang Yi* (皇矣) [2]. This marked the beginning of an English translation tradition that has continued for nearly three centuries [3]. Within this tradition, Arthur Waley's 1937 translation stands out for its departure from conventional arrangements. Waley reorganized the poems through the lens of cultural anthropology, offering a distinctive interpretive framework.

**Data availability statement:** The data underlying the results presented in the study are available from https://doi.org/10.6084/m9.figshare.29890838.v1.

**Funding:** This study was supported by [On the English Translation, Dissemination and Reception of *Book of Poetry* (*Shijing*) from the Perspective of Digital Humanities (22BYY039)]. The funders had no role in study design, data collection and analysis, decision to publish, or preparation of the manuscript.

**Competing interests:** The authors have declared that no competing interests exist.

Similarly, Ezra Pound's 1954 translation is notable for its imagist style, and it is often included in modern poetry anthologies as a creative work [3].

The concept of register is central to this study. It refers to "a variety associated with a particular situation of use (including particular communicative purposes)" [4, p. 6]. Within the framework of Multidimensional Analysis (MDA), register is operationalized as a functional language variation characterized by co-occurring linguistic features that reflect specific communicative goals and situational contexts. Register variation is therefore not limited to vocabulary choice but involves the interaction of multiple linguistic features working together to fulfill communicative purposes in specific contexts.

This study adopts Biber's MDA framework to investigate register variation in the English translations of *Shijing* by Arthur Waley and Ezra Pound. The suitability of MDA for this study stems from its ability to systematically identify and quantify the subtle differences in how translators' linguistic choices contribute to the broader communicative goals of the text. As poetry translation involves complex decisions that balance cultural, stylistic, and ideological considerations, MDA offers a rigorous framework to uncover these nuances. By applying MDA to the translations of *Shijing*, we can examine how features such as modality, interactivity, and information structure vary across the two translators' works, thus linking linguistic choices to broader thematic and ideological stances.

## 2. Previous literature on register variation in translation studies

Register variation, defined as functional language use shaped by context and communicative purpose, has long been a central concept in translation studies. Researchers have employed various approaches, particularly corpus-based methods, to examine how register shift across different translation genres. Among these, MDA, developed by Douglas Biber, has proven especially effective for systematically mapping register variation.

MDA has been widely applied to specialized domains to reveal patterns of linguistic variation. For instance, studies on legal texts [5,6], and business communication [7] demonstrate how co-occurring linguistic features can be quantitatively analyzed to understand register shifts. Similarly, its use in political texts [8] and social media texts [9] illustrates MDA's flexibility in capturing interactive and informational dimensions of language. Collectively, these applications underscore the value of MDA for producing rigorous, reproducible insights into translation register.

Despite the progress in technical and professional domains, literary translation, particularly poetry, remains relatively underexplored. Existing research has primarily focused on novels [10–12], leaving MDA's potential for uncovering the subtle linguistic features of poetry largely untapped. Poetry translation poses unique challenges due to its intricate cultural, stylistic, and linguistic elements. Register variations in translations of classical Chinese poetry, such as *Shijing*, have not yet been systematically investigated.

This gap highlights the need for applying MDA to explore how translators' choices, shaped by their ideological perspectives and poetic philosophies, influence register.

Such analysis is essential for understanding the interaction between linguistic features and the broader communicative goals in poetic translation. By bridging this gap, research can provide a more comprehensive account of how register functions across different genres and translation contexts.

## 3. Research design

### 3.1 Research questions

This research aims to address the following questions:

1. Do Arthur Waley's and Ezra Pound's translations of *Shijing* belong to the same register? If not, what are the differences?

2. What linguistic features account for the register differences between the two translations?

3. What factors influence these register differences?

### 3.2 Research objects

The research focuses on two translations. The first is *The Book of Songs*, translated by British sinologist Arthur Waley and published by Grove Press in 1996. The second is *The Classic Anthology Defined by Confucius*, translated by American poet Ezra Pound and published by Faber and Faber Ltd. in 1954.

To ensure comparability between the two versions, the titles in Waley's translation have been excluded, as Pound's translation omits them. All poems from *Shijing* are included in the analysis. The total character counts are 36,528 for Waley's translation and 44,170 for Pound's. These figures refer only to the translated text and exclude spaces as well as all paratextual material such as translator notes, commentary, or introductions.

### 3.3 Tools and methods

This study employs MDA and follows the approach outlined by Zhao [10] to quantify linguistic variation in translations. MDA, developed by Douglas Biber, applies factor analysis to identify sixty-seven linguistic features. It also calculates their frequency and standardized frequency per thousand words. These features are then analyzed to derive dimension scores, which represent different functional aspects of language use. There are seven dimensions in total. However, Dimension 7 is typically excluded in practical applications due to limited data and its lack of comparability with the other six dimensions [10].

Additionally, SPSS statistical analysis software was used to conduct significance tests on the dimension scores and Z-values calculated by MAT 1.3.3. Multiple linear regression was also applied to identify key linguistic features influencing variation across register dimensions. Finally, corpus retrieval from the bilingual parallel corpus of *Shijing* was conducted to provide supporting examples.

This combined approach is particularly well-suited to studying translation register, as it goes beyond impressionistic or qualitative judgments. By relying on statistically derived dimensions and rigorous testing, it enables systematic comparison of linguistic features across translations and reveals register-level patterns, such as differences in modality, interactivity, and informativity. They are directly relevant to our research questions, since these patterns illuminate how Waley's and Pound's translations of *Shijing* reflect distinct thematic orientations and communicative purposes.

## 4. Contrast of the register of English translations of *Shijing*

### 4.1 Overall register differences between Waley's and Pound's translations

To compare the six register dimensions, this study compiled Waley's and Pound's translations of *Shijing* into text files (TXT) using OCR and manual correction. The translations were then organized into 305 separate TXT documents,

corresponding to the 305 poems in the original *Shijing*. These 305 English texts from each translation were imported into MAT 1.3.3 for analysis. The average dimension values are presented in Table 1.

The results show that Waley's translation aligns with the "involved persuasion" register. This register is characterized by strong persuasive and argumentative features, which are typical of text types such as interviews and impromptu speeches. In contrast, Pound's translation falls under the "general narrative exposition" register. It emphasizes information delivery through narration, with typical text types including news reports, newspaper editorials, and science fiction novels.

To further compare the dimensional differences between the two translations, independent sample t-tests were conducted using SPSS, with the results presented in Table 2.

The results reveal significant differences between Waley's and Pound's translations in Dimensions 1, 2, and 6 ($p < 0.05$). Waley's translation exhibits stronger interactive features and on-line informational elaboration but weaker narrativity. In contrast, Pound's translation demonstrates higher informativeness and narrativity, but weaker on-line informational elaboration. Additionally, linguistic features in both translations were analyzed using MAT 1.3.3. Frequencies and standardized frequencies per thousand words were calculated for sixty-seven linguistic features.

Independent sample t-tests were then conducted on the features associated with Dimensions 1, 2, and 6, which showed significant differences. The results are presented in Table 3.

The results indicate that, among the twenty-four linguistic features in Dimension 1, seventeen show significant differences between Waley's and Pound's translations. This represents approximately 70.83% of all features in this dimension. In Dimension 2, six linguistic features were analyzed, and five exhibit significant differences, accounting for 83.3% of the total. In Dimension 6, two linguistic features were considered, with "Demonstratives" showing a significant difference.

## 4.2 Contrast of the register dimensions

To further explore the dimensions with significant differences, this study examines the relationship between linguistic features and these dimensions. Stepwise regression analysis was conducted on the linguistic features in Dimensions 1, 2, and 6.

**Table 1. Contrast of the Register Dimensions of Waley's and Pound's Translations of *Shijing*.**

| Dimension | | D1 | D2 | D3 | D4 | D5 | D6 | Register Attribution |
|---|---|---|---|---|---|---|---|---|
| Waley's Translation (N = 305) | M | 6.19 | 0.75 | 2.67 | −3.06 | −1.54 | 0.03 | Involved persuasion |
| | SD | 12.636 | 7.352 | 7.358 | 7.517 | 4.684 | 4.505 | |
| Pound's Translation (N = 305) | M | −4.34 | 2.53 | 1.73 | −2.18 | −2.14 | −1.66 | General narrative exposition |
| | SD | 11.625 | 11.161 | 7.140 | 6.129 | 3.265 | 2.743 | |

[a]M = mean; SD = standard deviation.

**Table 2. Contrast of Dimensional Differences between Waley's and Pound's Translations.**

| Dimension | Waley's Translation (N = 305) | | Pound's Translation (N = 305) | | Mean Deviation | t | p |
|---|---|---|---|---|---|---|---|
| | M | SD | M | SD | | | |
| **D1** | **6.19** | **12.636** | **−4.34** | **11.625** | **10.528** | **10.709** | **0.000** |
| **D2** | **0.75** | **7.352** | **2.53** | **11.161** | **1.113** | **−2.316** | **0.021** |
| D3 | 2.67 | 7.358 | 1.73 | 7.140 | 0.938 | 1.598 | 0.111 |
| D4 | −3.06 | 7.517 | −2.18 | 6.129 | 0.880 | −1.585 | 0.114 |
| D5 | −1.54 | 4.684 | −2.14 | 3.265 | 0.603 | 1.845 | 0.066 |
| **D6** | **0.03** | **4.505** | **−1.66** | **2.743** | **1.686** | **5.583** | **0.000** |

[a] Statistically significant differences are indicated in bold.

**Table 3. Contrast of the Linguistic Features Involved in Three Dimensions with Significant Differences.**

| Number | | Factor | Waley | Pound | t | p | Difference |
|---|---|---|---|---|---|---|---|
| 1 | D1 | **AMP (Amplifiers)** | **−0.15** | **−0.96** | **5.752** | **0.000** | **0.804** |
| 2 | | **ANDC (Independent Clause Coordination)** | **0.09** | **0.40** | **−1.984** | **0.048** | **0.314** |
| 3 | | AWL (Word length) | −1.06 | −0.97 | −1.252 | 0.211 | 0.087 |
| 4 | | CAUS (Causative Adverbial Subordinators) | −0.38 | −0.60 | 1.729 | 0.085 | 0.219 |
| 5 | | DEMP (Demonstrative Pronouns) | −0.35 | −0.47 | 1.166 | 0.244 | 0.120 |
| 6 | | **DPAR (Discourse Particles)** | **0.43** | **−0.20** | **3.254** | **0.001** | **0.630** |
| 7 | | **EMPH (Emphatics)** | **−0.19** | **−0.86** | **3.842** | **0.000** | **0.674** |
| 8 | | **FPP1 (First Person Pronouns)** | **0.97** | **0.17** | **7.038** | **0.000** | **0.792** |
| 9 | | INPR (Indefinite Pronouns) | 0.09 | −0.13 | 1.272 | 0.204 | 0.218 |
| 10 | | **JJ (Attributive Adjectives)** | **−0.04** | **0.72** | **−4.689** | **0.000** | **0.756** |
| 11 | | **NN (Total Other Nouns)** | **1.60** | **3.66** | **−13.359** | **0.000** | **2.053** |
| 12 | | **PIN (Total Prepositional Phrases)** | **−0.82** | **−0.49** | **−2.779** | **0.006** | **0.329** |
| 13 | | **PIT (Pronoun It)** | **0.56** | **−0.78** | **7.833** | **0.000** | **1.341** |
| 14 | | **POMD (Possibility Modals)** | **0.44** | **−0.30** | **2.942** | **0.003** | **0.753** |
| 15 | | **SPP2 (Second Person Pronouns)** | **0.45** | **0.03** | **2.854** | **0.004** | **0.419** |
| 16 | | TTR (Type-token Ratio) | −6.29 | −6.05 | −1.137 | 0.256 | 0.235 |
| 17 | | **VPRT (Present Tense)** | **0.04** | **−0.67** | **8.097** | **0.000** | **0.706** |
| 18 | | **XX0 (Analytic negation)** | **0.72** | **0.16** | **2.672** | **0.008** | **0.563** |
| 19 | | **BEMA (Be as Main Verb)** | **−0.23** | **−1.71** | **9.472** | **0.000** | **1.478** |
| 20 | | **CONT (Contractions)** | **−0.66** | **−0.33** | **−6.565** | **0.000** | **0.328** |
| 21 | | **PRIV (Private Verbs)** | **−0.86** | **−0.62** | **−2.131** | **0.033** | **0.234** |
| 22 | | PROD (Pro-verb Do) | −0.62 | −0.57 | −0.518 | 0.605 | 0.045 |
| 23 | | **STPR (Stranded Preposition)** | **−0.27** | **0.36** | **−2.430** | **0.016** | **0.628** |
| 24 | | THATD (Subordinator That Deletion) | −0.50 | −0.41 | −0.950 | 0.342 | 0.096 |
| 25 | D2 | **VBD (Past Tense)** | **−0.45** | **−0.65** | **2.389** | **0.017** | **0.199** |
| 26 | | **PEAS (Perfect Aspect)** | **−0.30** | **−1.25** | **6.332** | **0.000** | **0.952** |
| 27 | | PRESP (Present Participial Clauses) | 0.34 | 0.43 | −0.341 | 0.734 | 0.093 |
| 28 | | **PUBV (Public Verbs)** | **−0.87** | **−0.56** | **−2.018** | **0.044** | **0.311** |
| 29 | | **SYNE (Synthetic Negation)** | **1.56** | **4.74** | **−4.862** | **0.000** | **3.187** |
| 30 | | **TPP3 (Third Person Pronouns)** | **0.47** | **−0.19** | **5.809** | **0.000** | **0.668** |
| 48 | D6 | **DEMO (Demonstratives)** | **0.64** | **−0.93** | **6.259** | **0.000** | **1.567** |
| 49 | | THVC (That Verb Complements) | −0.61 | −0.73 | 0.749 | 0.454 | 0.119 |

**4.2.1 Contrast of informational and involved production.** Dimension 1 represents the contrast between informational and involved production. It reflects the interactivity and informativeness of texts. Higher values indicate greater interactivity and lower informativeness. This dimension is particularly relevant for poetic translation, as literary texts often require translators to navigate between conveying factual content and engaging readers emotionally. In other words, while some passages may demand precise transmission of information, others aim to evoke reader involvement, empathy, or aesthetic appreciation. Understanding Dimension 1 therefore helps reveal how translators balance these competing goals in their renderings of poetry.

This dimension includes thirty-four linguistic features, such as PRIV, TTR, VPRT, and AWL. For linguistic features that appear in multiple dimensions, those with higher loadings in other dimensions are excluded from the calculations for the current dimension [10]. Additionally, following Biber's [13] guidelines, features with loadings below 0.35 across all dimensions are excluded from the analysis.

In dimension 1, five features, such as COND and PLACE, have higher loadings on other dimensions and are therefore excluded. The feature "Present Participial WHIZ Deletions" is also excluded because its factor loadings are below 0.35 in all dimensions. Furthermore, as per Nini [14], features with Z-score means below 1 are excluded from the analysis.

After applying these criteria, only twenty-four linguistic features are retained for the calculation of scores in Dimension 1. To identify the features contributing to register differences, stepwise regression analysis was performed on these twenty-four features for both Waley's and Pound's translations. The results are presented in Tables 4 and 5.

The five linguistic features that most strongly contribute to the distinction between informational and involved production in Waley's translation are NN, POMD, AWL, PIN, and DPAR. Together, these features explain 63.5% of the variance in this dimension ($R^2 = 0.635$). The standardized regression equation is as follows: Dimension 1 (Waley's translation) = 1.742−2.379*NN + 1.013*POMD − 5.010*AWL − 2.597*PIN + 0.863*DPAR.

Among these features, POMD and DPAR have positive regression coefficients. In contrast, NN, AWL, and PIN have negative ones. This means that a higher frequency of possibility modals and discourse particles enhances textual interactivity. By contrast, longer average word length, greater use of nouns (excluding nominalizations and gerunds), and more prepositional phrases contribute to greater informativeness. The explanatory power of the first four linguistic features each exceeds 0.05 in terms of $R^2$ changes. DPAR accounts for an $R^2$ change of 0.042. This suggests that nouns (excluding nominalizations and gerunds), possibility modals, average word length, and prepositional phrases play a more significant role in shaping the interactivity of Waley's translation than discourse particles.

Further analysis of Table 3 shows that the p-values for AWL in both translations exceed 0.05. This indicates that AWL does not differ significantly between Waley's and Pound's versions. Therefore, the three linguistic features most responsible for the higher interactivity of Waley's translation are nouns (excluding nominalizations and gerunds), possibility modals, and prepositional phrases. Nouns function as primary carriers of referential meaning, with higher noun frequency

**Table 4. Parameters of the Stepwise Regression Models for the Top Five Variables Predicting the Scores of Dimension 1 of Waley's and Pound's Translations.**

| Model | R | R Square | Adjusted R Square | Std. Error of the Estimate |
|---|---|---|---|---|
| Waley's Translation | 0.797 | 0.635 | 0.629 | 7.69691 |
| Pound's Translation | 0.805 | 0.648 | 0.642 | 6.95792 |

**Table 5. Stepwise Regression Coefficients for Dimension 1 of Waley's and Pound's Translations.**

| Model | | Unstandardized Coefficients | | Standardized Coefficients | t | p |
|---|---|---|---|---|---|---|
| | | B | Standard Error | Beta | | |
| Waley's Translation | (Constant) | 1.742 | 0.969 | | 1.797 | 0.073 |
| | NN | −2.379 | 0.257 | −0.346 | −9.253 | 0.000 |
| | POMD | 1.013 | 0.125 | 0.289 | 8.088 | 0.000 |
| | AWL | −5.010 | 0.528 | −0.351 | −9.497 | 0.000 |
| | PIN | −2.597 | 0.306 | −0.305 | −8.488 | 0.000 |
| | DPAR | 0.863 | 0.147 | 0.210 | 5.861 | 0.000 |
| Pound's Translation | (Constant) | −4.240 | 1.246 | | −3.404 | 0.001 |
| | NN | −2.045 | 0.220 | −0.344 | −9.285 | 0.000 |
| | XXO | 1.725 | 0.183 | 0.334 | 9.408 | 0.000 |
| | FPP1 | 2.778 | 0.367 | 0.282 | 7.570 | 0.000 |
| | STPR | 0.672 | 0.095 | −0.246 | 7.045 | 0.000 |
| | TTR | −1.054 | 0.155 | −0.239 | −6.801 | 0.000 |

correlating with greater information density. Possibility modals introduce the author's viewpoint or inferential stance, actively engaging readers in interpretation and reinforcing interactivity. Prepositional phrases facilitate the integration of information within the text. Table 3 further demonstrates that, compared to Pound's translation, Waley's version uses significantly fewer nouns (excluding nominalizations and gerunds) and prepositional phrases but significantly more possibility modals. This combination results in a more involved register.

The five linguistic features that most significantly contribute to the score of Dimension 1 in Pound's translation are NN, XXO, FPP1, STPR, and TTR. Together, these features explain 64.8% of the variance in this dimension ($R^2 = 0.648$). The standardized regression equation is: Dimension 1 (Pound's translation) = −4.240−2.045*NN + 1.725*XXO + 2.778*FPP1 + 0.672*STPR − 1.054*TTR.

Among these, XXO, FPP1, and STPR have positive regression coefficients, while NN and TTR have negative ones. This indicates that a higher frequency of analytic negation, first-person pronouns, and stranded prepositions enhances textual interactivity. In contrast, a greater use of nouns (excluding nominalizations and gerunds) and a higher type-token ratio contribute to greater informativeness. The first three linguistic features (NN, XXO, and FPP1) show $R^2$ changes above 0.05. This suggests that, compared to stranded prepositions and type-token ratio, nouns (excluding nominalizations and gerunds), analytic negation, and first-person pronouns play a more significant role in shaping the informativeness of Pound's translation.

Table 3 reveals significant differences ($p < 0.05$) in the use of NN, XXO, and FPP1 between Pound's and Waley's translations. Therefore, nouns (excluding nominalizations and gerunds), analytic negation, and first-person pronouns are identified as the primary factors contributing to the higher interactivity of Waley's translation. Analytic negation (e.g., not) functions as an alternative to synthetic negation (e.g., no, neither). It often results in a more fragmented presentation of information and a reduction in informational density. First-person pronouns directly reference the speaker and are commonly used in interactive discourse [13].

**Example 1**

ST (Source Text): 有车邻邻, 有马白颠。未见君子, 寺人之令。阪有漆, 隰有栗。既见君子, 并坐鼓瑟。今者不乐, 逝者其耋。阪有桑, 隰有杨。既见君子, 并坐鼓簧。今者不乐, 逝者其亡。(国风·秦风·车邻)

WT (Waley's Translation): The coach-wheels crunch; There is one horse with a white forehead. **I** have not yet seen **my** lord; **I** am waiting till they send for **me**. On the hillside grows the lacquer-tree, On the lowlands the chestnut-tree. Now **I** have seen **my** lord; He sits opposite **me**, playing his zither, "If today **we** are not merry, In time to come **we** shall be too old." On the hillside grows the mulberry-tree, On the lowlands the willow. Now **I** have seen **my** lord; He sits opposite **me**, playing his reed-organ, "If today **we** are not merry, In time to come **we** shall be gone."

PT (Pound's Translation): Chariots, rank on rank with white-fronted horses; You'd see Milord? Eunuchs are bosses. Terebinth on the hill, chestnuts in valley; Once you're inside, there are lutes in each alley. Delight, delight and the long night coming. Mulberries on the crest, willows in marsh-land valley, drum-beat and shamisan, dally, dally, Death's up the alley.

This example shows that Waley's translation uses first-person pronouns more frequently than Pound's. The original poem reflects the life of the Qin ruler. Waley's use of first-person pronouns such as "we," "me," and "I" brings the emotional tone of the poem closer to the readers. By adopting a first-person perspective, Waley allows readers to directly experience the poet's emotions and inner thoughts. This approach enhances emotional resonance and builds an interactive connection between the poem and its audience. For instance, Waley translates "未见君子, 寺人之令" as "I have not yet seen my lord; I am waiting till they send for me." This rendering expresses the poet's anticipation from a personal viewpoint, immersing the reader in the poet's emotional state. Another example is the line "If today we are not merry, in time to come we shall be too old." Here, the use of "we" emphasizes the shared feelings and fate between the poet and the lord, creating a sense of unity.

In contrast, Pound's translation focuses on presenting information concisely. His aim is to retain the essential meaning of the original poem while simplifying its structure. This approach highlights Pound's emphasis on informativeness. This

can be observed in three key ways. First, he uses clear and concise sentence structures. For example, the line "Once you're inside, there are lutes in each alley" directly conveys the meaning of the original, avoiding excessive embellishment or verbosity. Second, he employs moderate reduction. Pound simplifies the original text but preserves its core meaning. For instance, "今者不乐, 逝者其亡" becomes "Death's up the alley." This succinctly expresses themes of death and loss while retaining emotional resonance. Third, he provides vivid depiction of scenes. For example, "有车邻邻, 有马白颠" is translated as "Chariots, rank on rank with white-fronted horses." This translation captures the imagery of rows of vehicles from the original text, enhancing the visual effect.

### Example 2

ST: 厌浥行露, 岂不夙夜？谓行多露。谁谓雀无角, 何以穿我屋？谁谓女无家, 何以速我狱？虽速我狱, 室家不足！谁谓鼠无牙, 何以穿我墉？谁谓女无家, 何以速我讼？虽速我讼, 亦不女从！(国风·召南·行露)

WT: The paths are drenched with dew. True, I said "Early in the night"; But I fear to walk in so much dew. Who **can** say that the sparrow has no beak? How else **could** it have pierced my roof? Who **can** say that you have no family? How else **could** you bring this suit? But though you bring a suit, Not all your friends and family will suffice. Who **can** say that the rat has no teeth? How else **could** it have pierced my wall? Who **can** say that you have no family? How else **could** you bring this plaint? But though you bring this plaint, All the same I will not marry you.

PT: "Dew in the morning, dew in the evening, Always too wet for a bridal day." The sparrow has no horn to bore a hole? Say you won't use your family pull! Not for the court and not for the bailiff, shall you make me a wife to play with. Toothless rat, nothing to gnaw with? And a whole family to go to law with? Take me to court, see what will come. Never, never, never will you drag me home.

This example illustrates Waley's greater reliance on possibility modals compared with Pound. The original poem presents a woman resisting an unwanted marriage. Waley highlights interactivity and speculation, aligning with an involved register, while Pound stresses assertion and independence, aligning with a more informational and declarative register.

Waley renders the woman's resistance through repeated modal constructions such as "Who can..." and "How else could..." These formulations replicate the ST's interrogative tone and create a speculative and argumentative atmosphere. This encourages readers to enter the speaker's reasoning process. By foregrounding uncertainty and possibility, Waley intensifies the woman's emotional struggle while also enhancing reader involvement. The final line, "all the same I will not marry you," keeps the refusal personal and emphatic, maintaining the interactive quality of the whole passage.

In contrast, Pound's translation reshapes the poem into categorical statements of defiance. Instead of speculative modals, he relies on emphatic negation and repetition, as in "Never, never, never will you drag me home." This rhetorical intensification conveys the heroine's determination in a strikingly modern idiom, portraying her as a bold and independent figure. Pound also compresses imagery into vivid, idiomatic expressions such as family pull and go to law, which shift the poem's focus from logical disputation to a more forceful assertion of autonomy. His approach emphasizes informativeness and narrativity over interactivity, presenting the woman's resistance as a clear, uncompromising declaration rather than an argumentative dialogue.

### Example 3

ST: 南有乔木, 不可休思。汉有游女, 不可求思。汉之广矣, 不可泳思。江之永矣, 不可方思。翘翘错薪, 言刈其楚。之子于归, 言秣其马。汉之广矣, 不可泳思。江之永矣, 不可方思。翘翘错薪, 言刈其蒌。之子于归。言秣其驹。汉之广矣, 不可泳思。江之永矣, 不可方思。(国风·周南·汉广)

WT: In the south is an upturning tree; One can**not** shelter under it. Beyond the Han a lady walks; One can**not** seek her. Oh, the Han it is so broad, One can**not** swim it, And the Kiang, it is so rough One can**not** boat it! Tall grows that tangle of brushwood; Let us lop the wild-thorn. Here comes a girl to be married; Let us feed her horses. Oh, the Han it is so broad, One can**not** swim it, And the Kiang, it is so rough One can**not** boat it! Tall grows that tangle of brushwood; Let us lop the mugwort. Here comes a girl to be married; Let us feed her ponies. Oh, the Han it is so broad, One can**not** swim it, And the Kiang, it is so rough One can**not** boat it.

PT: Tall trees there be in south countree, that give no shade to rest in, And by the Han there roam young maids, to whom there's no suggestin', that they should wade the Han by craft, or sail to Kiang's fount on a raft. I've piled high her kindling wood, and cut down thorns in plenty, to get the gal to go home with me. I've fed the horse she lent me. She will **not** wade the Han by craft, or sail to Kiang-fount on a raft. I have piled high the kindling wood, and cut down sandal trees, to get this girl to take a man, and raise the colts at ease. One does **not** wade the Han by craft, or reach the Kiang-fount on a raft.

This example illustrates that Waley's translation employs analytic negation more frequently than Pound's. In the original poem, the repeated use of "不可" conveys an unfulfilled desire for a distant woman.

Waley consistently renders these as "One cannot..." This uniform structure preserves the syntactic parallelism of the original, reinforcing its rhythm and cadence. The recurrence of "cannot" also intensifies the sense of impossibility and longing, drawing readers into the speaker's frustration and thereby heightening the interactivity of the translation. By systematically foregrounding negation, Waley constructs a mood of restrained yet persistent yearning, closely mirroring the affective resonance of the source text.

In contrast, Pound's translation uses a more narrative and descriptive strategy, rendering the actions and obstacles indirectly rather than explicitly negating them. For example, phrases such as "One does not wade the Han by craft" embed negation within broader contextual expressions, which softens the immediate impact of impossibility. This approach foregrounds imagery and the concrete actions surrounding the speaker's desire.

**4.2.2 Contrast of narrative and non-narrative concerns.** In the translation of classical poetry, narrativity captures how events, actions, or sequences are conveyed, reflecting the extent to which a poem tells a story or presents a narrative flow. Dimension 2 reflects the degree of narrativity in texts. Higher values indicate a greater degree of narrativity. This dimension includes ten linguistic features: VBD, TPP3, PEAS, PUBV, SYNE, PRESP, VPRT, JJ, WZPAST, and AWL. However, VPRT, JJ, WZPAST, and AWL exhibit higher loadings in other dimensions and are therefore excluded. Specifically, the loadings of VPRT on Dimensions 1 and 2 are 0.86 and −0.47, respectively. For JJ, the loadings on Dimensions 1 and 2 are −0.47 and −0.41. WZPAST has loadings of −0.34 on Dimension 2 and 0.40 on Dimension 5. AWL's loadings are −0.58 on Dimension 1 and −0.31 on Dimension 2.

Positive values indicate a positive correlation, while negative values indicate a negative one. In cross-dimensional comparisons, the absolute values of the loadings are considered. As a result, only six linguistic features remain for calculating the scores of Dimension 2. Stepwise regression analysis was then performed in SPSS on these six features for both Waley's and Pound's translations. The results are presented in Tables 6 and 7.

The five linguistic features that contribute most to the non-narrativity of Waley's translation are SYNE, PRESP, PEAS, TPP3, and PUBV. Together, they account for 97.6% of the variance in this dimension ($R^2 = 0.976$). The standardized regression equation is: Dimension 2 (Waley's translation) = −0.545 + 1.003*SYNE + 1.005*PRESP + 0.992*PEAS + 1.173*TPP3 + 0.999*PUBV.

All regression coefficients are positive. This means that a higher frequency of synthetic negation, present participial clauses, perfect aspect verbs, third-person pronouns, and public verbs corresponds to greater narrativity. The $R^2$ change for each feature exceeds 0.05, confirming their strong explanatory power for the high non-narrativity in Waley's translation. However, Table 3 shows no statistically significant difference in the use of PRESP between the two translations (p = 0.734 > 0.05). Therefore, the main linguistic features accounting for the stronger non-narrativity in Waley's translation are synthetic negation, perfect aspect verbs, third-person pronouns, and public verbs.

Synthetic negation enhances textual clarity by emphasizing specific characteristics, reducing ambiguity, and ensuring a more precise transmission of the author's intent. As its frequency increases, narrativity becomes more pronounced. Perfect aspect verbs indicate completed past actions, reinforcing temporal sequencing. Third-person pronouns refer to animate entities, typically humans, distinguishing referents from the speaker and addressee. Public verbs (e.g., admit, assert, declare, hint, report, say) signal indirect or reported speech [13].

 

Table 6. Parameters of the Stepwise Regression Models for the Top Five Variables Predicting the Scores of Dimension 2 of Waley's and Pound's Translations.

**Table 6. Parameters of the Stepwise Regression Models for the Top Five Variables Predicting the Scores of Dimension 2 of Waley's and Pound's Translations.**

| Model | R | R Square | Adjusted R Square | Std. Error of the Estimate |
|---|---|---|---|---|
| Waley's Translation | 0.988 | 0.976 | 0.975 | 1.15580 |
| Pound's Translation | 0.997 | 0.995 | 0.994 | 0.82964 |

**Table 7. Stepwise Regression Coefficients for Dimension 2 of Waley's and Pound's Translations.**

| Model | | Unstandardized Coefficients | | Standardized Coefficients | t | p |
|---|---|---|---|---|---|---|
| | | B | Standard Error | Beta | | |
| Waley's Translation | (Constant) | −0.545 | 0.079 | | −6.863 | 0.000 |
| | SYNE | 1.003 | 0.012 | 0.754 | 81.760 | 0.000 |
| | PRESP | 1.005 | 0.023 | 0.399 | 43.679 | 0.000 |
| | PEAS | 0.992 | 0.028 | 0.322 | 35.608 | 0.000 |
| | TPP3 | 1.173 | 0.040 | 0.268 | 29.319 | 0.000 |
| | PUBV | 0.999 | 0.038 | 0.240 | 26.401 | 0.000 |
| Pound's Translation | (Constant) | −0.444 | 0.076 | | −5.851 | 0.000 |
| | SYNE | 0.991 | 0.005 | 0.890 | 206.911 | 0.000 |
| | PRESP | 0.999 | 0.013 | 0.340 | 79.320 | 0.000 |
| | PUBV | 1.026 | 0.024 | 0.187 | 43.202 | 0.000 |
| | PEAS | 1.120 | 0.044 | 0.110 | 25.691 | 0.000 |
| | TPP3 | 1.005 | 0.044 | 0.099 | 22.672 | 0.000 |

Table 3 further reveals that Waley's translation employs synthetic negation and public verbs less frequently, but employs perfect aspect verbs and third-person pronouns more extensively. This pattern creates an apparent contradiction. Although Waley's translation shows lower narrativity on the surface, the extensive use of PEAS and TPP3 that typically enhance narrativity suggests a tendency toward a philosophical-interpretive mode, where storytelling is more impersonal and reflective.

The five linguistic features that contribute most to the higher narrativity in Pound's translation are SYNE, PRESP, PUBV, PEAS, and TPP3. Together, they account for 99.5% of the variance in this dimension ($R^2 = 0.995$). The standardized regression equation is: Dimension 2 (Pound's translation) = −0.444 + 0.991*SYNE + 0.999*PRESP + 1.026*PUBV + 1.120*PEAS + 1.005*TPP3.

Among these features, the $R^2$ change for synthetic negation (SYNE) and present participial clauses (PRESP) exceeds 0.05. This indicates that these two features have stronger explanatory power for Pound's heightened narrativity compared to public verbs, perfect aspect verbs, and third-person pronouns. However, Table 3 shows no statistically significant difference in the use of present participial clauses (PRESP) between Pound's and Waley's translations ($p = 0.734 > 0.05$). Therefore, synthetic negation emerges as the most influential factor in explaining Pound's higher narrativity. Further analysis in Table 3 confirms that Pound's translation employs synthetic negation significantly more often than Waley's. This frequent use reinforces the stronger narrative quality of Pound's translation.

**Example 4**

ST: 旱既大甚, 则不可沮。赫赫炎炎, 云我无所。大命近止, 靡瞻靡顾。群公先正, 则不我助。

WT: The drought is long and deep, It cannot be curtailed. It is parching, a burning heat; We have **no** place to escape, The hand of fate closes near, There is none to look to, none to care. The fonner ministers and their lords, Even they do not give us aid.

PT: The great drought! None can withstay it, It is impetuous fire against which I have **no** recourse. The great destiny draws to an end, we find **neither** awe **nor** shelter, **nor** do the pastoral lords of aforetime bring us their aid,

This example demonstrates that Pound's translation uses significantly more synthetic negation than Waley's. The original poem depicts King Xuan of the Zhou dynasty praying for rain.

Waley's translation remains faithful to the original while incorporating a strong philosophical dimension. For instance, in his rendering, the line "It is parching, a burning heat; We have no place to escape" conveys not only a literal depiction of drought but also a broader existential struggle. The imagery of "parching" and "burning heat" extends beyond physical suffering to suggest spiritual desolation. It resonates with the notion of a disrupted cosmic order. This dual-layered interpretation illustrates Waley's skill in embedding philosophical depth within a faithful representation of the original text.

In contrast, Pound's translation amplifies emotive intensity and dramatic expression with synthetic negation. For example, the sentence "we find neither awe nor shelter, nor do the pastoral lords of aforetime bring us their aid" employs multiple instances of negation. This repetition reinforces a profound sense of alienation, emphasizing the individual's isolation from both divine and earthly sources of support. This choice heightens the translation's dramatic tone, immersing the reader in an existential crisis. While the approach enhances emotional impact and broadens its appeal to modern audiences, it also risks diverging from the original poem's communal and cosmological themes. The Zhou dynasty's prayer for rain, which reflects collective suffering and a plea for cosmic harmony, becomes in Pound's version a deeply individualistic lament, an interpretation shaped more by his own literary sensibilities than by the ethos of the source text.

**4.2.3 Contrast of on-line informational elaboration.** Dimension 6 differentiates translations based on their degree of on-line information elaboration. Higher values indicate a more informational style, often produced under time constraints, like speeches. This dimension originally includes nine linguistic features: THAC, DEMO, TOBJ, THVC, STPR, EX, DEMP, WHOBJ, and PHC. However, several features are excluded due to cross-dimensional distribution or low statistical significance:

- EX and DEMP have loadings below 0.35 across all seven dimensions.

- STPR has a higher loading on dimension 1.

- WHOBJ and PHC load more heavily on dimension 3.

- The mean Z-scores of THAC and TOBJ are below 1.

As a result, only two linguistic features, DEMO and THVC, are retained for calculating Dimension 6 scores. Stepwise regression analysis was then conducted in SPSS on DEMO for both Waley's and Pound's translations. The results are presented in Tables 8 and 9.

The regression results show that the most significant linguistic feature explaining Dimension 6 scores in both translations is the use of demonstrative pronouns (DEMO). Demonstrative pronouns account for 76.0% of the score in Waley's translation ($R^2 = 0.760$) and 62.8% in Pound's translation ($R^2 = 0.628$). The standardized regression equations are: Dimension 6 (Waley's translation) = $-0.627 + 1.024$*DEMO; Dimension 6 (Pound's translation) = $-0.699 + 1.003$*DEMO.

The positive regression coefficients suggest that a higher frequency of demonstrative pronouns correlates with a more informational text. Since cohesion in unplanned informative discourse largely relies on demonstrative pronouns [13], their increased use strengthens textual cohesion. As shown in Table 3, it is evident that Waley's translation employs significantly more demonstrative pronouns than Pound's. This explains why Waley's translation exhibits stronger on-line information elaboration.

**Example 5**
ST: 有狐绥绥, 在彼淇梁。心之忧矣, 之子无裳。有狐绥绥, 在彼淇厉。心之忧矣, 之子无带。有狐绥绥, 在彼淇侧。心之忧矣, 之子无服。(国风·魏风·有狐)

**Table 8. Parameters of the Stepwise Regression Models for the Top Variable Predicting the Scores of Dimension 6 of Waley's and Pound's Translations.**

| Model | R | R Square | Adjusted R Square | Std. Error of the Estimate |
|---|---|---|---|---|
| Waley's Translation | 0.872 | 0.760 | 0.760 | 2.20906 |
| Pound's Translation | 0.792 | 0.628 | 0.626 | 1.67689 |

**Table 9. Stepwise Regression Coefficients for Dimension 6 of Waley's and Pound's Translations.**

| Model | | Unstandardized Coefficients | | Standardized Coefficients | t | p |
|---|---|---|---|---|---|---|
| | | B | Standard Error | Beta | | |
| Waley's Translation | (Constant) | −0.627 | 0.128 | | −4.888 | 0.000 |
| | DEMO | 1.024 | 0.033 | 0.872 | 31.008 | 0.000 |
| Pound's Translation | (Constant) | −0.699 | 0.105 | | −6.658 | 0.000 |
| | DEMO | 1.033 | 0.046 | 0.792 | 22.594 | 0.000 |

WT: There is a fox dragging along By that dam on the Ch'i. Oh, my heart is sad; **That** man of mine has no robe. There is a fox dragging along By **that** ford on the Ch'i. Oh, my heart is sad; **That** man of mine has no belt. There is a fox dragging along By **that** side of the Ch'i. Oh, my heart is sad; **That** man of mine has no coat.

PT: K'i dam, prowls fox, a heart 's to hurt and someone's there has got no skirt. By the K'I's deep on the prowl; got no belt on, bless my soul. Tangle-fox by K'i bank tall: who says: got no clothes at all?

This example demonstrates that Waley's translation uses demonstrative pronouns more frequently than Pound's. The original poem, written from the perspective of a woman, laments her husband's destitution and lack of clothing as he wanders [15].

Waley's translation prominently features the demonstrative pronoun "that", whcih is used in two distinct contexts: to refer to objects and to denote locations. First, "that" is used to specify "之子" in the original poem, as seen in Waley's translation: "That man of mine." This usage clarifies the reference and enhances the explicitness of the relationship between the characters. It reduces the reader's cognitive load and improving the translation's readability. Second, "that" is used to indicate specific locations, such as "that dam," "that ford," and "that side," corresponding to "淇梁," "淇厉," and "淇侧" in the original. This strategy localizes the poem's geographic setting, allowing readers to visualize the scenes more intuitively. It also strengthens the spatial coherence of the translation.

In contrast, Pound's translation prioritizes vivid visual imagery over demonstrative pronouns. He emphasizes the portrayal of the fox and the depiction of the environment. Pound conveys the fox's mysterious and agile nature through dynamic verbs like "prowl" and "tangle," creating a sense of movement and enigma. Additionally, Pound's translation intensifies the visual impact of the landscape by using phrases like "By K'I's deep" and "by K'i bank tall," evoking the depth of the Qi River and the steepness of its banks. This interplay between the fox's elusive movements and the detailed environmental descriptions creates a heightened sense of mystery and vitality. Moreover, Pound introduces a conversational tone by including expressions like "bless my soul," which injects a conversational tone, further enlivening the translation.

## 5. Discussion on the causes of register differences

To further analyze the register differences between Waley's and Pound's translations, this study explores the ideological, poetic, and patronage factors influencing their composition, framed within the context of manipulation theory. According to this framework, translation is not a neutral transfer of meaning but a process shaped by ideological, poetic, and patronage constraints [16]. These constraints determine translators' linguistic choices, which in turn shape register. By examining

Waley's and Pound's works through this lens, we can see how their translations are not only linguistic acts but also interventions situated within wider cultural and political contexts.

## 5.1 Contrast of ideology

Arthur Waley and Ezra Pound, two influential literary translators, approached *Shijing* with different ideologies, shaping their translation methodologies and the register of their respective works.

Waley initially treated *Shijing* with some skepticism. He argued that, of the three hundred and five poems, only a minority would engage a modern readership. He believed most of the poems focused on war or love and that their beauty could not be replicated in English [17]. In the preface to *The Temple and Other Poems* he contends that these folk-songs were originally assembled for political purposes and later interpreted by Confucian commentators as ethical texts. Love-poems were frequently allegorized as relations between ruler and minister [18]. Waley's political views aligned with those of the Bloomsbury Group, and his anti-imperialist stance became more pronounced after World War I, particularly in advocating for the return of looted Chinese artworks to China [19]. His encounter with the work of scholars like Hu Shi and Gu Jiegang, who applied Western methods to study Chinese history, further influenced his views, leading him to see *Shijing* as no longer sacred [20]. After reading Marcel Granet's anthropological analysis of *Shijing*, Waley redefined it as a reflection of primitive Chinese social life, translating the title as *Book of Songs* rather than *Book of Poetry*. This anthropological and anti-imperialist orientation prompted Waley to foreground social context and human agency in his English translations. Linguistically, this tendency is reflected in his use of analytical negation, first-person pronouns, modal verbs, and demonstratives, which increase interactivity and elaborative texture.

By contrast, Pound construed *Shijing* primarily as a vehicle for moral and educational instruction. He regarded Confucianism as a corrective to what he perceived as the moral decay of Western civilization. Translating *Shijing* was not just a scholarly endeavor but a politically motivated intervention to restore social order and moral responsibility in the West [21]. This interpretive stance carries the risk, as noted by Said [22], of reproducing an Orientalist dynamic. In this framework, the "East" is reshaped into an idealized moral archetype, serving as a mirror for Western self-reflection and as a backdrop for projecting civilizational superiority. Pound's selective emphasis on odes praising rulers reflects his authoritarian leanings. Figures such as King Wen are cast as emblems of centralized power, with phrases like "King Wen's law is our right" reinforcing his belief in authoritarian rule [23]. This moralizing, politically inflected approach contrasts sharply with Waley's more anthropological and socially contextualized reading of *Shijing*. Pound's convictions shaped the register of his translation, producing discourse marked by high informativeness and narrativity. His reliance on dense nominal and prepositional constructions foregrounds substantive content while minimizing embellishment. At the same time, his use of synthetic negation and public verbs generates a tightly structured, authoritative narrative. This narrative structure underscores the significance of Confucian principles, presenting them as a coherent ideological framework to address the moral challenges facing Western society.

In sum, Waley's anthropological and anti-imperialist stance produced a register oriented toward interpersonal engagement and contextual elaboration. This is realized through interactive markers such as first-person pronouns, modals, and analytic negation, as well as demonstratives that anchor the reader in social circumstances. MDA characterizes this pattern as "involved persuasion." By contrast, Pound's politically instrumentalized and morally didactic orientation produced a register classified as "general narrative exposition." This register is characterized by high nominal density, prepositional framing, synthetic negation, and the use of public verbs that foreground informational content, narrative coherence, and authoritative moral assertions. From the perspective of manipulation theory, these ideological commitments function as constraints that directly governed linguistic form. Waley manipulated the text to foreground social context and resist imperialist readings, while Pound manipulated it to construct a moralizing discourse that aligned Confucian ideals with his authoritarian vision for the West.

## 5.2 Contrast of the poetics

Arthur Waley and Ezra Pound represent two distinct translation philosophies whose poetics systematically conditioned their linguistic choices and hence their register profiles. This section explores how Waley's focus on fidelity, clarity, and cultural respect contrasts with Pound's emphasis on energy, emotion, and precision in their renditions of *Shijing*.

Waley, influenced by the modernist poetry movement, departed from Victorian conventions and rejected English poetic metrics in translating classical Chinese poetry. His poetics emphasize fidelity to the source, attention to detail, and respect for the cultural and historical context of the poems. Notably, Waley's translations are non-Orientalist, avoiding the Euro-centric embellishments common in earlier scholarship [24]. He maintained a literalist approach, aiming for "literal translation, not free translation" [25, p. 33], and his work fits Xu Yuanchong's [26] category of literalist translators. For instance, in his rendering of the poem 采薇, he translates the line "靡室靡家, 玁狁之故. 不遑启居, 玁狁之故" (No house, no home, nomadic tribes' reason. Not at ease, settle, nomadic tribes' reason) as "We have no house, no home Because of the Hsien-yun. We cannot rest or bide Because of the Hsien-yun." This translation exemplifies key aspects of Waley's poetics and illustrates how his choices shape the translation's register. He preserves semantic fidelity by translating "玁狁" (a northern ethnic group in ancient China and an enemy of the Zhou dynasty) as "Hsien-yun," retaining the ethnonym rather than generalizing it as "enemies" or "nomads," thus conveying the poem's socio-political context. The added subject "we," absent in the original, foregrounds human agency and the collective experience, anchoring the narrative in the speakers' perspective and enhancing reader engagement. The translation also maintains syntactic clarity and parallelism, with both couplets echoing the original rhythm and structural symmetry. Repeating "Because of the Hsien-yun" preserves the cause-and-effect logic, while the simple sentence structures improve readability. Together, Waley's literal fidelity, contextual embedding, and syntactic clarity create an "involved persuasion" register, marked by interactivity, elaboration, and attention to social and historical context.

Ezra Pound's poetics, defined by energy, emotion, and precision, offer a distinct approach to translation. For Pound, energy in poetry captures the power of tradition, race consciousness, and cultural association. In *How to Read*, Pound categorizes poetic energy into three forms: Melopoeia (sound and rhythm), Phanopoeia (imagery), and Logopoeia (intellectual atmosphere). Pound's use of Logopoeia revitalizes *Shijing* by emphasizing the nuanced interplay between words, transcending their direct meanings. In his translation of 采薇, for instance, he translates "采薇采薇, 薇亦作止" as "Pick a fern, pick a fern, ferns are high," creating a dynamic flow that mirrors the original's energy. This contrasts with Waley's more literal translation, which adds narrative elements. Emotion is central to Pound's poetics, guiding both form and content [27]. He argues that emotions should shape word choice, infusing poetry with authenticity. In his 采薇 translation, the phrase "Home', I'll say: home, the year's gone by" invites readers to feel the poet's emotions, blending homesickness with the passage of time. Pound's personal emotional connection deepens the poem's emotional resonance, especially during his internment. In addition, Pound also emphasizes precision, using words that directly convey meaning while eliminating excess [27]. His translation of 采薇 is more concise than Waley's, by omitting the subject and simplifying syntax, focusing on the action itself, thus creating a direct and impactful translation. His deliberate omission of lines like "忧心烈烈..." enhances the poem's rhythm and thematic clarity, highlighting precision as key to emotional depth.

In sum, Waley's literalist, culture-respecting poetics yield translations that elaborate social context and foreground interpersonal stance (e.g., demonstratives). This generates an "involved persuasion" register, characterized by interactivity and elaboration. By contrast, Pound's energetic, emotion-driven, and precisionist poetics produce compressed, image-centric lines with high nominal density and narrativity, corresponding to the "general narrative exposition" register. These divergent poetics exemplify how aesthetic priorities function as a mode of textual manipulation. Translators not only render meaning but also reshape the source text's style to align with their own literary agendas and the expectations of their readership. This aesthetic manipulation is inseparable from the register distinctions observed in their works.

## 5.3  Contrast of the patrons

In the publication of their translations, both Arthur Waley and Ezra Pound were shaped to some extent by their patrons. However, patronage did not significantly affect the register of their translations.

Before translating *Shijing*, Waley had already gained financial independence and scholarly recognition through numerous publications. His translations of Bai Juyi's poems and pre-Tang poetry in *Bulletin of the School of Oriental Studies* earned praise from critic Arthur Clutton-Brock in *The Times* supplement. This endorsement led Constantinople Publishing Company to release *One Hundred and Seventy Chinese Poems*, which achieved great success and established Waley's reputation [28]. Subsequent works, including *More Translations from the Chinese* (1919), *The Temple and Other Poems* (1923), *The Tale of Genji* (1925–1933), and *The Way and Its Power* (1934), further consolidated his standing in sinology and poetry translation. By 1929, Waley retired from the British Museum, gaining financial independence and the freedom to pursue his studies [19]. The first Waley's edition of *Shijing* appeared in 1937, published by George Allen & Unwin Ltd., a leading twentieth-century British press. Its mass-oriented, avant-garde philosophy complemented Waley's clear and accessible style. Prior collaborations with the publisher ensured a strong relationship, and the translation received favorable circulation. Since its debut, Waley's *Shijing* has been acclaimed by scholars and general readers alike. Over time, it underwent revisions and multiple reprints, producing more than ten editions and establishing a "monument in the history of Chinese poetry translation" [29, p. 115]. The 1937 edition remains the first public translation, reflecting Waley's original vision [30]. Subsequent editions include the second and third by George Allen & Unwin (1954, 1969), the 1987 Grove Press edition with a foreword by Stephen Owen, and the 1996 edition edited by Joseph R. Allen. Allen's edition reinstated fifteen political lament poems, reorganized the sequence into "Feng, Ya, Song," added titles, and replaced proprietary names with Pinyin, while retaining Waley's style.

In publishing Ezra Pound's *Shijing*, Achilles Chih-tung Fang, a Harvard scholar, played a crucial sponsorship role while Pound was imprisoned at St. Elizabeths Hospital. Pound viewed the 305 poems as meaningful both in writing and rhythm, aiming for a version with English-Chinese parallel texts and phonetic annotations. He corresponded frequently with Willis Hawley, owner of Los Angeles Chinese Bookstore, James Laughlin, president of New Directions Publishing, and printer Dudley Kimball to achieve this trilingual alignment. In 1951, Fang took over aligning the text and annotating characters from Kimball [31]. Fang initially tried to publish Pound's *Shijing* as a trilingual edition via New Directions, but Laughlin doubted its market potential and cited high typesetting costs [32]. Fang then approached Harvard University Press, which only wanted Pound's English translation. After Fang's mediation, in 1953, Pound and Harvard University Press agreed to first publish a commercially viable English translation, followed by an academic English-Chinese parallel edition with character annotations [33]. In the preface, Fang noted: "A volume containing a sound key to the 305 Odes (transcription of each syllable), along with the Chinese text in seal script and the English translation, will be published" [34, p. xiv]. Through this arrangement, Pound's *Shijing* was finally published in 1954. Fang's role was pivotal. He assisted with typesetting, proofreading, and acted as a vital liaison between the publisher and Pound, especially during the translator's imprisonment.

In sum, compared to ideology and poetic concepts, patronage had limited influence on the register formation in Waley and Pound's translations. While patrons provided financial support and facilitated publication, they affected the translators' ideological and poetic choices only indirectly. The register were shaped primarily by the translators' own concepts of poetry translation. Thus, patronage enabled publication but was not a determining factor in the creation of translation register.

## Author contributions

**Resources:** Guangwei Li.

**Supervision:** Guangwei Li.

**Validation:** Guangwei Li.

**Writing – original draft:** Baohu Li.

**Writing – review & editing:** Baohu Li.

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
