## [Decision Letter · Decision Letter 0]

15 Jul 2025

Dear Dr. Li,

Thank you for submitting your manuscript to PLOS ONE. After careful consideration, we feel that it has merit but does not fully meet PLOS ONE’s publication criteria as it currently stands. Therefore, we invite you to submit a revised version of the manuscript that addresses the points raised during the review process.

Thank you for your submission to PLOS ONE. After a thorough evaluation of the manuscript titled "A Multi-dimensional Analysis of Register Variations in the English Translations of Shijing", and a careful review of the three peer reviewer reports, I am recommending **major revisions** before further consideration.

<h3 data-end="519" data-start="483">Required Changes for Acceptance:</h3>

**Terminology Alignment with MDA Framework** : The manuscript must revise its use of the term "register" to align with Douglas Biber’s definition in the context of Multi-Dimensional Analysis (MDA). Reviewer 1 rightly highlights conceptual inconsistencies that undermine the study's methodological foundation. The term must be applied accurately, and the authors should avoid conflating MDA with cluster analysis or stylistic variation outside the MDA framework.**Clarify Theoretical Justification and Method** : The manuscript must explicitly clarify why MDA was selected as the method, and how it has been correctly applied to poetry translation. The theoretical basis (Biber’s framework) must be better explained and directly linked to the research questions and linguistic features analyzed.**Data Availability Compliance** : Although a GitHub link is provided, the authors must ensure that all linguistic data supporting their statistical claims are fully accessible and described in the Data Availability Statement, in line with PLOS ONE’s policy.**Language and APA Style Compliance** : Substantial editorial revisions are required to improve clarity and consistency. This includes correcting grammatical issues, simplifying complex sentence structures, and ensuring APA-style in-text citations and references (e.g., changing “Wang et al., 2021: 67” to “Wang et al., 2021, p. 67”).

<h3 data-end="1996" data-start="1921">Recommended Changes (not required for acceptance but strongly advised):</h3>

**Expand Literature Review with Greater Thematic Coherence** : The current literature review reads as a list. Reorganize thematically (e.g., by domain or method) and integrate more directly with the research objective, particularly in the underrepresented area of poetry translation using MDA.**Strengthen Integration of Textual Examples** : While statistical results are sound, the manuscript would benefit from more illustrative examples that make the differences between Waley and Pound’s styles tangible. Several reviewers recommend including quotations that exemplify the highlighted features.**Discussion Section Improvements** : Reviewers suggest summarizing the ideological and poetic differences at the end of each subsection (5.1, 5.2) to clearly link them back to the observed register distinctions. Consider adding a brief reference to postcolonial or Orientalist frameworks to enrich the ideological discussion, particularly regarding Pound’s interpretation.

<h3 data-end="3019" data-start="2978">Conflicting Reviewer Recommendations:</h3>

Reviewer 1 recommended rejection, primarily due to conceptual misalignment and terminological misuse.Reviewer 2 recommended minor revisions, acknowledging the technical rigor of the data analysis but suggesting expansion of theoretical and contextual grounding.Based on my own evaluation, I find merit in the study's ambition and quantitative approach, but I agree with Reviewer 1 that fundamental theoretical and definitional revisions are needed. Therefore, I am not supporting rejection at this stage, but a **major revision** that directly addresses these core issues.

<h3 data-end="3617" data-start="3602">Conclusion:</h3>

In its current form, the manuscript does not yet meet PLOS ONE’s publication criteria, particularly with respect to conceptual clarity, methodological rigor, and data transparency. However, with significant revision, it has the potential to make a valuable contribution to cross-cultural stylistic analysis and corpus-based translation studies. I encourage the authors to revise carefully in light of the above feedback and the detailed reviewer comments.

We look forward to receiving your revised manuscript.

Kind regards,

Ramandeep Kaur

Academic Editor

PLOS ONE

Journal Requirements: 

 [On the English Translation, Dissemination and Reception of Book of Poetry (Shijing) from the Perspective of Digital Humanities  (22BYY039)]. 

3. Thank you for uploading your study's underlying data set. Unfortunately, the repository you have noted in your Data Availability statement does not qualify as an acceptable data repository according to PLOS's standards.

Additional Editor Comments:

Thank you for submitting your manuscript titled "A Multi-dimensional Analysis of Register Variations in the English Translations of Shijing." The topic is compelling and the study attempts an important application of Biber’s Multidimensional Analysis (MDA) to classical poetry translation—an area with limited prior research.

However, based on the reviewers' feedback and a careful reading of the manuscript, several key concerns must be addressed before the manuscript can be considered for publication.

1. Clarification of Theoretical Concepts

The manuscript frequently uses the term "register" in ways that are not aligned with Douglas Biber’s definition within the MDA framework. In MDA, registers are empirically derived groupings based on co-occurring linguistic features in specific situational contexts. The analysis needs to better reflect this definition.

Additionally, the discussion on cluster analysis appears to conflate MDA with MD typology analysis. Please clarify your methodological approach and ensure it is consistent with Biber’s model.

2. Literature Review Expansion and Depth

The review of prior work on register variation in translation is broad but lacks coherence. Please restructure it to clearly show thematic connections and relevance to poetry translation and MDA.

Specific references to studies on Arthur Waley’s and Ezra Pound’s translation styles would greatly strengthen the conceptual grounding.

3. Data Transparency

Reviewer 2 has noted issues with data availability. While a GitHub repository is listed, please ensure that all raw data used for analysis—including the linguistic feature counts and text files—is clearly described in the manuscript and accessible via the repository.

4. Stylistic and Structural Revisions

Numerous instances of long and dense sentences reduce readability. Please revise for clarity, conciseness, and standard academic tone.

Correct the formatting of in-text citations to comply with APA style (e.g., replace “Wang et al., 2021: 67” with “Wang et al., 2021, p. 67”).

Fix typographical errors such as “Disscusion” → “Discussion,” and ensure that all tables (e.g., the missing Table 5.3) are correctly numbered and referenced.

5. Anchoring Analysis with Textual Examples

While the statistical results are thorough, the manuscript would benefit from more direct and interpretive textual examples. For instance, quote full sentences that illustrate the features like pronoun use or negation forms discussed in the statistical sections.

6. Discussion of Translator Ideology and Poetics

The discussion section is conceptually rich but could benefit from clearer summaries at the end of each subsection (e.g., ideology, poetics) to explicitly connect these elements to the observed register differences.

Consider briefly engaging with alternative perspectives, such as Orientalism or postcolonial theory, especially when discussing Pound’s ideological motivations.

Reviewers' comments:

Reviewer's Responses to Questions

**Comments to the Author**

1. Is the manuscript technically sound, and do the data support the conclusions?

Reviewer #1: No

Reviewer #2: Yes

Reviewer #3: Partly

2. Has the statistical analysis been performed appropriately and rigorously?

Reviewer #1: No

Reviewer #2: Yes

Reviewer #3: Yes

3. Have the authors made all data underlying the findings in their manuscript fully available?

Reviewer #1: No

Reviewer #2: No

Reviewer #3: Yes

4. Is the manuscript presented in an intelligible fashion and written in standard English?

Reviewer #1: No

Reviewer #2: Yes

Reviewer #3: No

Reviewer #1: General comments

The paper shows a misunderstanding of key concepts in Multi-Dimensional Analysis. The term register is repeatedly misused throughout the text, diverging from Biber’s definition, where 'register' refers to a variety of language defined by a situational context and empirically identified through co-occurring linguistic features. The supposed ‘registers’ compared in the study do not correspond to those identified in MD Analysis, and the mention of cluster analysis confuses MD Analysis with MD text typology analysis. Moreover, Section 2 provides an inadequate review of relevant literature, and the research questions do not refer to the theoretical framework the study claims to adopt. Finally, the discussion section abandons the MD framework, which weakens conceptual coherence.

Specific comments

'variations'

=>

'variation,' in the singular, throughout.

The term "register" refers to the suitability of vocabulary and sentence structure for a particular style (Wang and Ding 1987: 414), representing a cluster of linguistic features that tend to co-occur more frequently than would be expected by chance (Halliday, 1988: 162).

=>

You need to use a definition of register from Biber, which is different from these.

Comparing the registers of Waley's and Pound's translations

=>

This use of the term 'register' does not correspond to the usage of the term in MD Analysis.

The advancements in natural language processing (NLP) have made corpus-based methods a key tool in poetry research.

=>

MD Analysis is a corpus linguistic approach, not an NLP one. This statement needs to be revised.

the appropriateness of language in specific contexts

=>

this is not the concept of register in MD Analysis

Section 2

=>

The overview of previous work is poorly presented.

Research questions

=>

These RQs do not make sense given the improper use of the term 'register,' as noted.

Tools and methods

=>

MDA itself does not employ cluster analysis, which is used for text typology.

Table 1

=>

The labels under the 'Register' column are not registers, in the regular MDA sense of the term.

Heading: Contrast of the register dimensions

=>

Again, improper use of the term 'register' in the MDA tradition.

Heading: Section 5. Disscussion [sic]

=>

This section moves away from MD Analysis entirely.

Reviewer #2: 1. Add to the literature review on the writing styles of Waley and Ezra.

2. Add to the literature review on the criteria used to analyze the text and why it has been chosen.

3. Page 11: as shown in table 5.3. There is no table with this number.

4. Shed light on why this research is important and what it adds to the literature.

Reviewer #3: The study offers an innovative comparative analysis of Arthur Waley’s and Ezra Pound’s translations of Shijing using Biber’s MDA framework. The contrast between “involved persuasion” and “general narrative exposition” registers is an original and insightful contribution to both translation studies and register analysis. However, several issues limit its scholarly impact in its current form:

• Theoretical Framing Needs Depth: Although the manuscript references manipulation theory, it does not engage meaningfully with this or any other theoretical lens (e.g., postcolonialism, Orientalism). A more explicit framework would ground the discussion more critically.

• Overreliance on Statistical Description: While the tables are detailed, their interpretation often lacks depth. For example, real textual excerpts from the translations are provided, but they could be more systematically tied back to the quantitative findings to demonstrate how linguistic features manifest in translation style.

• Literature Review is Overly Descriptive: The related works section reads like a list. It would be more useful to synthesize findings from related research and highlight gaps this study addresses.

• Writing and Structure Require Revision: There are numerous typographical and grammatical issues throughout the manuscript. Additionally, some sections are repetitive or lack clear transitions between ideas.

• Balance Between Translators: The discussion of Pound’s translation and ideology is more detailed than Waley’s, which causes an imbalance in analytical depth. Equalizing the treatment would strengthen the comparative nature of the study.

Therefore, my recommendations are:

1. Revising the manuscript for clarity, grammar, and citation style.

2. Adding more explicit theoretical engagement, particularly with manipulation theory and postcolonial perspectives.

3. Strengthening connections between quantitative data and textual analysis.

4. Balancing the depth of discussion between the two translators.

**Do you want your identity to be public for this peer review?** For information about this choice, including consent withdrawal, please see our Privacy Policy

Reviewer #1: No

Reviewer #2: No

Reviewer #3: No

---

## [Author Response · Author response to Decision Letter 1]

21 Aug 2025

We have carefully considered all comments from the reviewers and the editor. Detailed responses to each comment are provided in the document “Response to Reviewers,” which indicate the corresponding revisions made to the manuscript. All changes are highlighted in the document “Revised Manuscript with Track Changes.”

---

## [Editor Report · Decision Letter 1]

1 Sep 2025

Multidimensional Analysis of Register Variation in English Translations of Shijing

PONE-D-25-26290R1

Dear Dr. Li,

We’re pleased to inform you that your manuscript has been judged scientifically suitable for publication and will be formally accepted for publication once it meets all outstanding technical requirements.

Kind regards,

Ramandeep Kaur

Academic Editor

PLOS ONE

Additional Editor Comments (optional):

Thank you for carefully and thoroughly revising your manuscript in response to the reviewers’ and editor’s feedback. The revised version represents a significant improvement in clarity, rigor, and overall presentation.

Your use of Biber’s MDA framework is now conceptually accurate, with terminology clearly aligned to the original definitions.

The theoretical justification for applying MDA to poetry translation is well articulated, and the methods are explained in sufficient detail.

The data availability statement and Figshare repository ensure that all underlying data are accessible and replicable.

The literature review has been reorganized with stronger thematic coherence, and the inclusion of multiple textual examples makes the statistical contrasts more tangible.

The discussion section is more robust, clearly linking ideology and poetics to register distinctions, while also situating the study within broader socio-cultural contexts.

These revisions collectively address all of the major concerns raised during the review process. We are pleased to accept your article for publication in PLOS ONE. Congratulations on this contribution, which we believe will be of value to both translation studies and corpus-based stylistics.
---

## [Editor Report · Acceptance letter]

PONE-D-25-26290R1

PLOS ONE

Dear Dr. Li,

I'm pleased to inform you that your manuscript has been deemed suitable for publication in PLOS ONE. Congratulations! Your manuscript is now being handed over to our production team.

Kind regards,

on behalf of

Dr. Ramandeep Kaur

Academic Editor

PLOS ONE